# Treatment Outcomes of Langerhans Cell Histiocytosis: A Retrospective Study

**DOI:** 10.3390/medicina57040356

**Published:** 2021-04-07

**Authors:** Kazuhiko Hashimoto, Shunji Nishimura, Naoki Sakata, Masami Inoue, Akihisa Sawada, Masao Akagi

**Affiliations:** 1Department of Orthopedic Surgery, Kindai University Hospital, Osaka-Sayama City, Osaka 589-8511, Japan; shunnisi@med.kindai.ac.jp (S.N.); makagi@med.kindai.ac.jp (M.A.); 2Department of Pediatrics, Kindai University Hospital, Osaka-Sayama City, Osaka 589-8511, Japan; nsakata@med.kindai.ac.jp; 3Department of Hematology/Oncology, Osaka Women’s and Children’s Hospital, Izumi City, Osaka 594-1101, Japan; pedino@wch.opho.jp (M.I.); asawada@wch.opho.jp (A.S.)

**Keywords:** Langerhans cells, histiocytosis, outcomes

## Abstract

*Background and Objectives*: Langerhans cell histiocytosis (LCH) is a rare disease characterized by the infiltration of one or more organs by Langerhans cell-like dendritic cells. LCH often involves the bone, and its clinical evidence is limited. The purpose of this study is to report on the treatment of LCH at our institution and to add to the evidence for LCH. *Materials and Methods*: We reviewed six cases of LCH treated in our hospital between November 2005 and February 2016. Patient age at the first visit, sex, site of origin, symptoms, image tools used for diagnosis, biopsy site, complications, treatment, and final clinical outcome were evaluated. The median follow-up period was 41 months. *Results*: The median patient age at the first visit was 13.5 years. Three male and three female individuals were enrolled. Multiple lesions were observed in five cases, and a solitary lesion was observed in one case. Pain was the chief complaint in five cases. Radiography was the most commonly used imaging tool. Bone scintigraphy or magnetic resonance imaging and positron emission tomography-computed tomography were also used to diagnose systematic LCH. Biopsy of the femur was performed in two cases, and biopsy of the tibia, lumbar vertebrae, rib, and radius was performed in one case each. Regarding comorbidities, one case of hepatitis B and one case of autism were observed. Chemotherapy was initiated in two patients. The other four patients were observed naturally. Continuous disease-free survival was observed in five patients. One patient remained alive but not without disease during the final follow-up examination. *Conclusion*: LCH should be diagnosed as early as possible to treat it appropriately.

## 1. Introduction

Langerhans cell histiocytosis (LCH) is a rare clonal neoplastic proliferation of myeloid dendritic cells driven by mutations in the mitogen-activated protein kinase pathway [1,2]. On immunohistology, LCH is positive for CD1a and S100 antigens [3]. LCH tends to occur in young adults and children [4,5], and it can be unifocal within a single (SS) or a multi-system (MS) [1]. LCH-SS involves the bones, such as the skull, femur, vertebrae, pelvis, ribs, and mandible; lymph nodes; skin; and the lungs [1]. In contrast, the skin, bone, liver, spleen, and bone marrow are mainly involved in LCH-MS [1]. Patients with LCH-SS are usually older children or adults with painful lytic lesions eroding the bone cortex [1]. Solitary lesions in other sites are present as masses [1]. Patients with LCH-MS are usually young children with multiple or sequential destructive bone lesions, often associated with adjacent soft tissue masses [1]. The involvement of high-risk organs, such as the liver and spleen, is a poor prognostic sign [1,6].

Because of the lack of clinical evidence, the pathogenesis of LCH remains unclear.

Here, we describe the features and outcomes of clinical cases to provide information on LCH.

## 2. Patients and Methods

We retrospectively reviewed six cases of LCH treated at our hospital between November 2005 and February 2016.

Patient data, such as age at the first visit, sex, site of origin, symptoms, image tools used for diagnosis, biopsy site, complications, treatment, and the final clinical outcome were reviewed.

The median follow-up period was 41 months (range: 13–73 months).

## 3. Results

The patients’ characteristics are presented in Table 1. The median patient age at the first visit was 13.5 years (range: 2–60). The study included three male and three female patients. There were five cases involving multiple lesions and one case involving a single lesion. Trunk lesions, such as those in the rib or lumbar vertebrae, were involved in all three cases of adults aged >20 years old. However, pediatric cases involved the extremities. No patients had oral lesions. Five patients were symptomatic at the time of their first visit, and one was asymptomatic. Among the symptomatic cases, pain was the most common complaint. Claudication was a common complaint in patients with limb onset. The most commonly used imaging tool for diagnosis was radiography, which was used in all cases. Gallium scintigraphy, magnetic resonance imaging (MRI), and positron emission tomography-computed tomography (PET-CT) were used in four, three, and two cases, respectively. In all cases, multiple images were used for diagnosis. Biopsies were performed on the tibia, lumbar vertebrae, ribs, and radius in one case each, and the femur was involved in two cases. Hepatitis B and autism were the observed comorbidities in one case each. Four patients were followed up with imaging, and two patients were treated with chemotherapy.

The final clinical outcome in five patients was continuous disease-free. One patient remained alive but not without disease. We present the case of a 60-year-old male patient with LCH-MS who was treated with chemotherapy, as he had a MS-type disease. Since his prostate-specific antigen level was 4.2 ng/mL, which was higher than the normal range, an abdominal magnetic resonance imaging (MRI) examination was performed. The MRI revealed the presence of a lesion, suspected to be a metastatic bone tumor, in the lumbar vertebrae, and the patient was referred to our hospital (Figure 1a,b). Bone scintigraphy showed accumulation of the radiotracer in the right fourth rib (Figure 1c). Prostate biopsy did not reveal any malignant tumors. PET-CT was performed based on the diagnosis of cancer of unknown primary origin. Multiple accumulation and osteolytic changes were observed in the fourth and fifth lumbar vertebrae, sacrum, bilateral ilium, right fourth rib, fifth thoracic vertebra, left neck, and left submandibular gland (Figure 2a–f, respectively). He underwent bone biopsy of the fifth lumbar vertebra. The pathological findings showed multinucleated giant cells in the connective tissue, eosinophils, and lymphocytes infiltrating the surrounding lesion, and mononuclear cells with mild nuclear atypia (Figure 3a). Immunohistochemistry was positive for langerin and S100 (Figure 3b,c, respectively) and negative for CD-1a (Figure 3d). The diagnosis was confirmed based on the findings. In total, nine chemotherapy courses with vinblastine (6 mg/m^2^), prednisolone (2 mg/kg), 6-mercaptopurine (1.5 mg/kg), and methotrexate (20 mg/m^2^) were administered. He was in remission after chemotherapy. Approximately 4.5 years after chemotherapy, recurrence has not been observed.

## 4. Discussion

LCH is a rare disease and there is little coherent evidence regarding this issue. Therefore, we reviewed and summarized the clinical features of LCH from a case that we have encountered. The current study provides evidence for the pathogenesis of LCH.

The prevalence of LCH is 0.5–0.9/100,000 in children aged <15 years and 0.1/100,000 in people aged >15 years [7]. The median age of patients at LCH diagnosis is 3.5 years (most commonly before 1 year of age) among children [2]. LCH most commonly occurs in adults in their 30s and 40s [7]. In a previous work, the male-to-female ratio was found to be 1.2:1, [1]. In this study, the results were bimodal, especially the patients aged ≤3 years and ≥20 years. The male-to-female ratio was 1:1. 

LCH can be divided into two types: the SS and MS types. LCH-SS involves the bone (skull, femur, vertebrae, pelvis, ribs, and mandible), lymph nodes, skin, and lungs [1,7]. LCH can occur in any bone in the body, except for the hands and feet, and it is unifocal in 75% of cases [2]. Moreover, >50% of the isolated bone lesions were associated with children aged <5 years [7]. In children and adults, the skull is the most common site of involvement [8]. In addition, 25% of LCH cases involved skin lesions [9]. There is an overhang of lesions in the soft tissues around the bone, accompanied by pain and swelling [10,11]. In this study, all pediatric cases involved the bones of the extremities, and all adult cases involved the bones of the trunk. No skin lesions were observed. Pain, swelling, and claudication associated with pain were also observed in five out of six patients. There was one rare case of radial development.

A systemic survey of lesions is essential for LCH because the treatment options and prognosis depend on the extent of the disease [12,13]. The guidelines recommend the performance of chest and skeletal radiographic screening [14]. However, a previous study reported that the detection rate, as reported by the only radiographic survey of the skeleton to date, was approximately 50–60%, which was unsatisfactory [15]. Whole-body MRI was also significantly more useful in detecting lesions in LCH than radiography, as previously described [15]. A PET-CT scan was also beneficial for the whole-body survey of LCH lesions, although the exposure dose was relatively high [16,17,18]. In this study, radiographic examination of the skeleton at the local site of pain and swelling symptoms helped in the search for primary lesions or to locate primary lesions. In addition, the survey for systemic lesions was mainly performed using bone scintigraphy and PET-CT. These findings suggested that it was necessary to examine the lesions comprehensively using various imaging tests and clinical findings.

Pathological findings are the most important tools in the definitive diagnosis of LCH [7]. Identification of Langerhans cells may be the key for diagnosis [1]. Langerhans cells are oval in shape and recognized by the grooved, folded, indented, or lobed nuclei and their chromatin, inconspicuous nucleoli, and thin nuclear membranes [1]. Immunostaining is also useful for identifying Langerhans cells [1]. They express CD1a, CD207, S100, CD68, and human leukocyte antigen – DR isotype (HLA-DR) [1]. The main differential diagnoses include osteomyelitis, chondroblastoma, and multiple myeloma [1,19]. The definitive diagnosis in this study was based on pathological diagnosis using immunostaining. Although we did not experience any serious complications during biopsy in this study, a minimally invasive and safe biopsy site and method should be selected for LCH-MS.

The localized form (LCH-SS) has been reported to achieve remission spontaneously [20]. Therefore, surgical treatment should be avoided for the single type and no systemic chemotherapy is needed [2]. However, LCH-MS should be treated with systemic chemotherapy [2]. Randomized studies have been conducted in pediatric patients, and chemotherapy with methotrexate, vinblastine, 6-mercaptopurine, and etoposide was reported to be effective. These medications are also used in adults [21,22]. Recently, other protocols, such as using cladribine and cytarabine, have been tolerated and applied [23]. In this study, we used 6-mercaptopurine, methotrexate, and prednisolone for MS-type LCH in adults. Cladribine and cytarabine were also used for MS-type LCH in children. Chemotherapy was administered because the LCH was of the MS type.

LCH-SS has a favorable prognosis, with a survival rate of approximately 100% [2]. Moreover, the 5-year recurrence rate was <20% [2]. The recurrence rate of LCH-MS was ≤50%, as previously described [10,21]. The 5-year overall survival rates of LCH-SS and LCH-MS without risk of organ involvement (bone marrow, liver, and spleen) were 100% and 98%, respectively [2]. Meanwhile, the survival rate in patients with at-risk organs was found to be ≤77% [2,10,21]. Although there were no cases involving high-risk organs in this study and there were no deaths or recurrences in all cases, careful follow-up examination is needed.

Nevertheless, this study had some limitations. First, the number of cases included in the present study was small. Therefore, we could not perform statistical analyses. Second, this study had a retrospective design, and the indications for chemotherapy varied depending on the treatment time. We would like to continue our research in the future by adding more cases so that we will be able to perform statistical analyses after matching the indications for chemotherapy induction.

## 5. Conclusions

In the current study, the clinical outcomes of LCH were favorable. Immediate diagnosis of LCH using imaging tools or biopsy is needed to avoid unnecessary treatment. An early decision should be made regarding the initiation of chemotherapy, in which case, it should be started immediately.

## Figures and Tables

**Figure 1 medicina-57-00356-f001:**
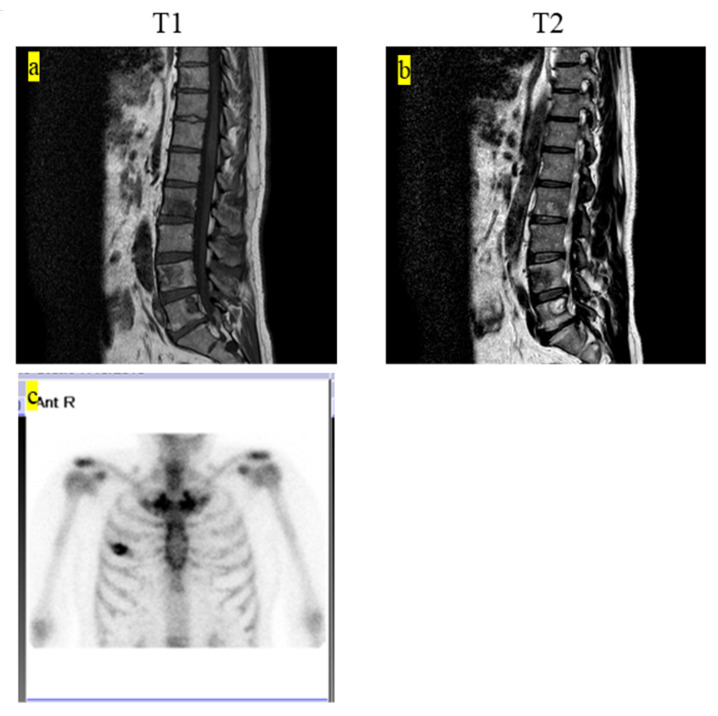
(**a**) T_1_-weighted MRI showing multiple low-intensity lesions. (**b**) T2-weighted MRI showing multiple low- and high-mosaic intensity lesions. (**c**) Bone scintigraphy shows the accumulation of the radiotracer on the right fourth rib. MRI, magnetic resonance imaging.

**Figure 2 medicina-57-00356-f002:**
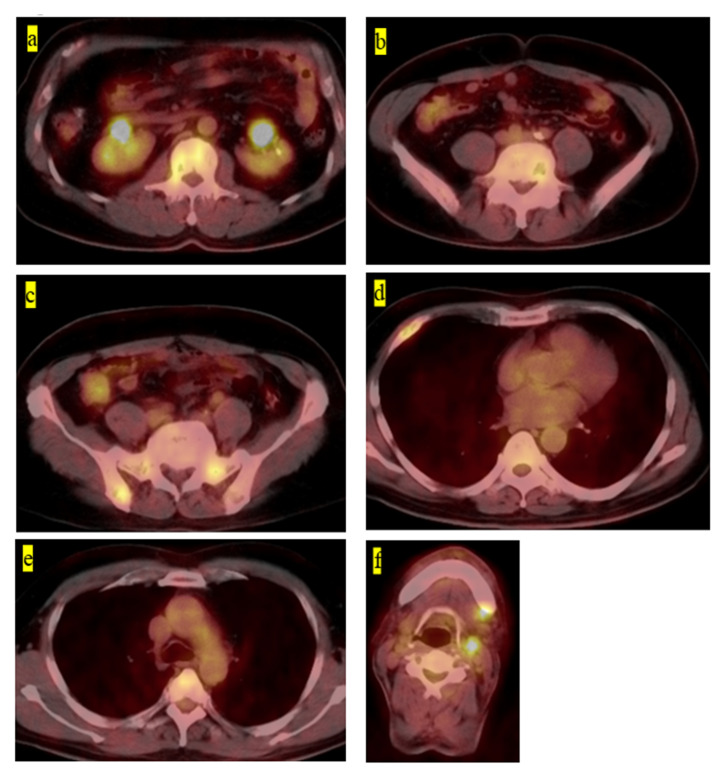
(**a**) Fluorodeoxyglucose-positron emission tomography–computed tomography image showing the accumulation on the fourth lumbar vertebra, (**b**) fifth lumbar vertebra, (**c**) right pelvic bone, (**d**) right fourth rib, (**e**) fifth thoracic vertebra, and (**f**) left neck and left submandibular gland.

**Figure 3 medicina-57-00356-f003:**
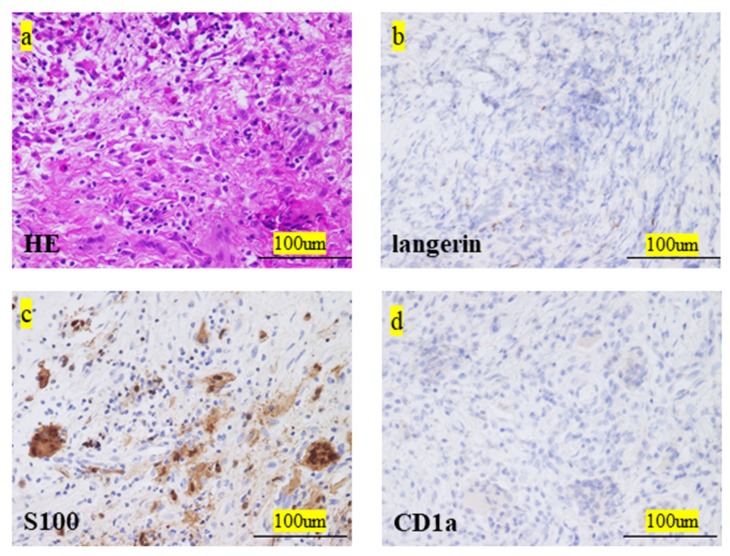
(**a**) Histological examination of H-E stained sections. (**b**) Immunohistochemical staining examination for langerin (**c**) and S-100 was positive. (**d**) Immunohistochemical staining examination for CD-1a was negative. H-E, hematoxylin–eosin.

**Table 1 medicina-57-00356-t001:** Clinical features of Langerhans cell histiocytosis (LCH) cases in the current study.

Patient No.	Age (Years)	Sex	Site	Imaging Tool for Diagnosis	Symptom	Comorbidity	Biopsy Site	Treatment	Outcome	Follow-Up Period (Months)
1	24	F	Rt tibia, left seventh rib	Radiography, scintigraphy, PET-CT	Pain and swelling	None	Tibia	Spontaneous regression	CDF	13
2	60	M	Second, third, fourth, fifth lumbar, Th10, rt fourth rib, sacrum, pelvic	Radiography, MRI, PET-CT	Without symptoms	Hepatitis B	Fifth lumbar	Chemotherapy (6-mercaptopurine 110 mg, MTX 2.5 mg, PSL 5 mg)	CDF	63
3	30	F	Rt fourth rib	Radiography, scintigraphy, CT, MRI, contrast-enhanced MRI	Pain	None	Rt fourth rib	Natural observation	AWD	4
4	2	F	Femur, eighth thoracic vertebra	Radiography, MRI	Pain and claudication	None	Femur	Spontaneous regression	CDF	27
5	2	M	Femur, temporal bone, anterior mediastinum	Radiography, scintigraphy	Pain and claudication	Autism	Femur	Chemotherapy (cladribine 5 mg/m^2^ and cytarabine 500 mg/m^2^)	CDF	55
6	3	M	Elbow	Radiography, Scintigraphy	Pain	None	Radius	Spontaneous regression	CDF	73

CDF, continuous disease-free; AWD, alive with disease; MRI, magnetic resonance imaging; PET-CT, positron emission tomography–computed tomography; LCH, Langerhans cell histiocytosis; PSL, prednisolone; MTX: methotrexate; F, female; M, male.

## Data Availability

The datasets used and/or analyzed during the current study are available from the corresponding author on reasonable request.

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
