# Peer review of "Treatment Outcomes of Langerhans Cell Histiocytosis: A Retrospective Study"

_medicina, 2021, doi:10.3390/medicina57040356_

Round 1
Reviewer 1 Report
The article aims at raising awareness amongst professionals so that, in the presence of a disease such as the Langerhans cell histiocytosis (LCH), a diagnostic protocol using imaging and biopsy is applied in order to plan the most correct and adequate treatment.
The article is sufficiently interesting and well enough articulated to comply with its objective of sensitizing professionals.
However, the work does not provide any specific advance towards the current knowledge and does not address this important long-standing question with smart experiments. Hence its contribution to research is rather limited.
Additionally, and more specifically:
- The sample of cases under review is extremely limited;
- It is not clear whether the age of the patients indicated in the table 1 was that reported at the beginning of the period of the study or at the end of the period of follow-up;
- In which years have the patients been treated? This would be important to know because in the meantime treatments have changed;
- The title of the article does not correspond to the content, particularly as someone would expect to read of a study conducted during the period of observance while, in reality, the clinical study was carried out retroactively;
- The conclusions are extremely succinct and poorly elaborated. Moreover, they are obvious because they are the results of a quite limited study conducted retroactively;
- It seems that there were no cases with impacts in the oral cavity; was that investigated at all?
- In the abstract, some inconsistencies in reporting the data should be better revised. For instance the sentence "Chemotherapy was initiated in two patients. The other five patients were observed naturally" is confusing because it would seem that the sample cases are 7 and not 6.
Author Response
Reviewer 1
The authors would like to thank the reviewer for his/her constructive critique to improve the manuscript. We have made every effort to address the issues raised and to respond to all comments. The revisions are indicated in blue font in the revised manuscript. Please find next a detailed, point-by-point response to the reviewer's comments.
Comment
The article aims at raising awareness amongst professionals so that, in the presence of a disease such as the Langerhans cell histiocytosis (LCH), a diagnostic protocol using imaging and biopsy is applied in order to plan the most correct and adequate treatment.
The article is sufficiently interesting and well enough articulated to comply with its objective of sensitizing professionals.
However, the work does not provide any specific advance towards the current knowledge and does not address this important long-standing question with smart experiments. Hence its contribution to research is rather limited.
Additionally, and more specifically:
Comment 1
The sample of cases under review is extremely limited;
Response
We would like to thank the reviewer for the comment. Indeed, as the reviewer pointed out, the number of patients was small. Nevertheless, please note that it was the total number of patients with LCH that we have treated at our institution so far.
Although the number was small, we believe that our work has provided some evidence.
In the revised manuscript, we have stated that the small number of patients was a limitation of our study. We would like to continue examining such cases and report the data of more patients with LCH in the future.
Please note that we have added the following part at the end of the Discussion section in the revised manuscript: “Nevertheless, this study had some limitations. First, the number of cases included in the present study was small. Therefore, we could not perform statistical analyses. Second, this study had a retrospective design, and the indications for chemotherapy varied depending on the treatment time. We would like to continue our research in the future by adding more cases so that we would be able to perform statistical analyses after matching the indications for chemotherapy induction.” (Lines 173–178)
Comment 2
It is not clear whether the age of the patients indicated in the table 1 was that reported at the beginning of the period of the study or at the end of the period of follow-up;
In which years have the patients been treated? This would be important to know because in the meantime treatments have changed;
Response:
The ages of the patients presented in Table 1 correspond to their ages at the time of their first visit to our department. We have provided this information in the revised manuscript as follows: “The median age at the first visit was 13.5 years (range, 2–60).” (Lines 55–56)
Moreover, please note that the treatment duration of each case was as follows (biopsied):
Case 1: November 2005 (Just biopsy)
Case 2: From August 2015 to November 2015 (biopsy and chemotherapy)
Case 3: May 2007 (Just biopsy)
Case 4: November 2014 (Just biopsy)
Case 5: From July 2015 to February 2016 (biopsy and chemotherapy)
Case 6: November 2014 (Just biopsy)
Based on the above information, we have corrected the treatment period in the revised manuscript as follows: “We retrospectively reviewed six cases of LCH treated at our hospital between November 2005 and February 2016.” (Lines 48–49)
Comment 3
The title of the article does not correspond to the content, particularly as someone would expect to read of a study conducted during the period of observance while, in reality, the clinical study was carried out retroactively;
Response:
We would like to thank the reviewer for the comment
We agree with the reviewer that the title of the paper did not correspond to its content.
Therefore, we have revised the title as follows: “Treatment Outcomes of Langerhans Cell Histiocytosis: A Retrospective Study.”
Comment 4
The conclusions are extremely succinct and poorly elaborated. Moreover, they are obvious because they are the results of a quite limited study conducted retroactively;
Response:
We would like to thank the reviewer for pointing this out.
We agree with the reviewer that the conclusions were extremely succinct and poorly elaborated. Moreover, we should have stated that the retrospective design of the study was a limitation. Please note that we discussed this issue as a limitation in the revised manuscript as follows: “Nevertheless, this study had some limitations. First, the number of cases included in the present study was small. Therefore, we could not perform statistical analyses. Second, this study had a retrospective design, and the indications for chemotherapy varied depending on the treatment time. We would like to continue our research in the future by adding more cases so that we would be able to perform statistical analyses after matching the indications for chemotherapy induction.” (Lines 173–178)
Moreover, we have added the following sentence in the Conclusion section: “An early decision should be made regarding the initiation of chemotherapy, in which case, it should be started immediately.” (Lines 181–183)
Comment 5
It seems that there were no cases with impacts in the oral cavity; was that investigated at all?
Response:
We would like to thank the reviewer for the question.
Please note that no oral lesions were found in any of the patients included in this study.
We have provided this information in the revised manuscript as follows: “No patients had oral lesions.” (Lines 60)
Comment 6
In the abstract, some inconsistencies in reporting the data should be better revised. For instance the sentence "Chemotherapy was initiated in two patients. The other five patients were observed naturally" is confusing because it would seem that the sample cases are 7 and not 6.
Response:
We would like to apologize to the reviewer for the mistake. Please note that we have corrected this mistake in the revised manuscript.
Reviewer 2 Report
Overall a well written and interesting manuscript on Langerhans Cell Histiocytosis. I have just a few comments/questions:
line 58: 'among the asymtomatic cases, pain was the most common complaint'. I presume you mean 'symptomatic cases'
line 65: 'Hepatitis B and autism were the associated complications...'. Were these complications, or comorbidities?
line 66..., two were treated with chemotherapy. What was the reason to treat these cases with chemotherapy? Was this decision based on the extension of disease/multisystem involvement?
Author Response
Reviewer2
Comment
Overall a well written and interesting manuscript on Langerhans Cell Histiocytosis. I have just a few comments/questions:
Response
The authors would like to thank the reviewer for his/her constructive critique to improve the manuscript. We have made every effort to address the issues raised and to respond to all comments. The revisions are indicated in blue font in the revised manuscript. Please, find next a detailed, point-by-point response to the reviewer's comments.
Comment 1
line 58: 'among the asymtomatic cases, pain was the most common complaint'. I presume you mean 'symptomatic cases'
Response
We would like to apologize to the reviewer for the mistake. Indeed, we aimed to state “symptomatic cases.” We have corrected this mistake in the revised manuscript (Line 61).
Comment 2
line 65: 'Hepatitis B and autism were the associated complications...'. Were these complications, or comorbidities?
Response
We would like to thank the reviewer for the question. Please note that hepatitis B and autism were comorbidities. We have provided this information in the revised manuscript (Lines 25–26, 68–69; Table 1).
Comment 3
line 66..., two were treated with chemotherapy. What was the reason to treat these cases with chemotherapy? Was this decision based on the extension of disease/multisystem involvement?
Response
We would like to thank the reviewer for the suggestion.
In both cases, chemotherapy was administered because of the multiple forms of the disease.
We have provided this information in the revised manuscript as follows: “We present the case of a 60-year-old male patient with LCH-MS who was treated with chemotherapy, as he had a MS-type disease.” (Lines 76–79)
“Chemotherapy was administered because the LCH was of MS-type.” (Line 163-164)
Round 2
Reviewer 1 Report
Having the authors made all the suggested changes and having also addressed all the comments I made during the first review, I confirm that the paper has improved significantly and should be accepted in the present form.